# Socioeconomic inequality in knowledge about HIV/AIDS over time in Ethiopia: A population-based study

**Aklilu Endalamaw** [1,2]*, **Charles F. Gilks**[1], **Fentie Ambaw**[2], **Resham B. Khatri**[1,3], **Yibeltal Assefa**[1]

**1** School of Public Health, The University of Queensland, Brisbane, Australia, **2** College of Medicine and Health Sciences, Bahir Dar University, Bahir Dar, Ethiopia, **3** Health Social Science and Development Research Institute, Kathmandu, Nepal

\* yaklilu12@gmail.com

**Data Availability Statement:** All data underlying the findings are provided in the submitted

## Abstract

Socioeconomic inequality in comprehensive knowledge about HIV/AIDS can hinder progress towards ending the epidemic threat of this disease. To address the knowledge gap, it is essential to investigate inequality in HIV/AIDS services. This study aimed to investigate socioeconomic inequality, identify contributors, and analyze the trends in inequality in comprehensive knowledge about HIV/AIDS among adults in Ethiopia. A cross-sectional study was conducted using 2005, 2011, and 2016 population-based health survey data. The sample size was 18,818 in 2005, 29,264 in 2011, and 27,261 in 2016. Socioeconomic inequality in comprehensive knowledge about HIV/AIDS was quantified by using a concentration curve and index. Subsequently, the decomposition of the concentration index was conducted using generalised linear regression with a logit link function to quantify covariates' contribution to wealth-based inequality. The Erreygers' concentration index was 0.251, 0.239, and 0.201 in 2005, 2011, and 2016, respectively. Watching television (24.2%), household wealth rank (21.4%), ever having been tested for HIV (15.3%), and education status (14.3%) took the significant share of socioeconomic inequality. The percentage contribution of watching television increased from 4.3% in 2005 to 24.2% in 2016. The household wealth rank contribution increased from 14.6% in 2005 to 21.38% in 2016. Education status contribution decreased from 16.2% to 14.3%. The percentage contribution of listening to the radio decreased from 16.9% in 2005 to -2.4% in 2016. The percentage contribution of residence decreased from 7.8% in 2005 to -0.5% in 2016. This study shows comprehensive knowledge about HIV/AIDS was concentrated among individuals with a higher socioeconomic status. Socioeconomic-related inequality in comprehensive knowledge about HIV/AIDS is woven deeply in Ethiopia, though this disparity has been decreased minimally. A combination of individual and public health approaches entangled in a societal system are crucial remedies for the general population and disadvantaged groups. This requires comprehensive interventions according to the primary health care approach.

manuscript. Additionally, we utilized data from the Demographic Health Survey (DHS) Program, as mentioned in the manuscript. The Ethiopian demographic health survey data that we obtained from DHS was acquired through an official request made to DHS. To access the DHS Program, please visit their website at https://dhsprogram.com/.

**Funding:** The authors received no specific funding for this work.

**Competing interests:** The authors have declared that no competing interests exist.

## Introduction

HIV/AIDS is a global public health problem that necessitates comprehensive HIV prevention services, including knowledge-related services, and integration of HIV services into routine care [1, 2]. Knowledge-building services are associated with capacity building, social marketing, advocacy, peer- or school-based education, and community mobilization [3–5]. In addition to comprehensive HIV interventions, these services have demonstrated efficiency and effectiveness in promoting positive behavioral change [6–8], encouraging health-seeking behavior [9, 10], preventing new HIV infections, and reducing the burden of disability-adjusted life years by approximately one million [3, 4]. As a result, they are included in health care parameters and can be measured accordingly.

The Joint United Nations Program for HIV/AIDS (UNAIDS), social organizations, and health policies have utilized knowledge as a monitoring indicator in the HIV/AIDS response [11–15]. Knowledge-related specific indicators are awareness about HIV/AIDS, knowing that consistent condom use and having one uninfected sexual partner prevent HIV, responding that a healthy-looking person can have HIV/AIDS, and rejecting the misconceptions that a person can get HIV from a mosquito bite and by sharing a meal with people living with HIV [16]. These indicators have guided numerous studies in estimating comprehensive knowledge and have revealed varying levels of knowledge across populations [17, 18]. Furthermore, evidence suggests that assessing the extent and sources of socioeconomic inequality exposes hidden disparities in knowledge about HIV/AIDS [19, 20].

Investigating inequality in knowledge about HIV/AIDS offers several benefits. First, it helps to develop and provide tailored interventions by reaching individuals with low service coverage [21]. Second, it enhances individuals' health care seeking behvaior [22]. Third, it promotes social justice in the community by highlighting the fair distribution of resources and opportunities among different groups [23]. Fourth, it reveals policies and practices that perpetuate disparities based on social determinants of health. This information can inform policymakers and programme officers about the necessity of targeted services to achieve universal health coverage (UHC) [24].

Universal Health Coverage is a Sustainable Development Goal 3.8 target, aiming to provide health care to everyone worldwide without financial jeopardy. Many countries have plans in place to achieve this goal [25]. For example, Ethiopia plans to increase comprehensive knowledge coverage to 90% among key and priority populations by 2025 [26]. In Ethiopia, certain studies estimated comprehensive knowledge about HIV/AIDS among young and reproductive women without considering the extent of contributors to socioeconomic inequality and changes over time [27, 28]. However, understanding how knowledge is changing within different social groups can be crucial for effective planning for future resources, leadership, and financial requirements. Social groups play a significant role as contributors, often referred to as 'the causes of the causes', and identifying the sources of socioeconomic inequality can support efforts to address these disparities [29]. For example, a study conducted in Malawi revealed that comprehensive knowledge about HIV/AIDS was concentrated among the richest group, with education being the primary contributor to socioeconomic inequality [20]. Previous studies, including those conducted in Ethiopia, have not attempted to identify sources and trends of inequality in knowledge about HIV/AIDS.

The study was conducted in Ethiopia, one of Africa's most populated countries, with a population of about 126 million [30]; of which, 21.3% are urban residents [31]. Ethiopia ranks 153rd out of 167 countries in the Prosperity Index [32]. In Ethiopia, HIV/AIDS is the 6th and 13th leading cause of mortality among females and males, respectively [33]. Women are at a higher risk of HIV infection, with approximately 0.36 million women and 0.22 million men living with HIV in Ethiopia [34].

This study contributes to the literature gap regarding inequality in comprehensive knowledge about HIV/AIDS. The objective of this study was to assess the status and contributing factors of socioeconomic inequality in knowledge about HIV/AIDS over time in Ethiopia.

## Methods

### Study setting, design, sample size and data source

The current study is reported based on the Strengthening the Reporting of Observational Studies in Epidemiology Statement: guidelines for reporting observational studies as presented in S1 Checklist. The study represents the Ethiopian population aged 15 to 49 years old adults. Ethiopia is one of the east African countries. We used a cross-sectional design, utilizing population-based data from the Ethiopian Demographic and Health Surveys (EDHS) conducted from 2005 to 2016. Participants were selected using multistage sampling techniques. EDHS conducted two-stage cluster sampling procedures, in which samples are stratified, clustered, and selected in two stages. The base for stratification was urban and rural, which are clustered in nine regions and two city administrations [16, 35, 36]. The final sample size was 18,818 in 2005, 29,264 in 2011, and 27,261 in 2016. Fig 1 displays the flow chart of the sample for the analysis. Enumeration areas were listed from November 2004 to January 2005, September 2010 to January 2011, and September to December 2015 for the 2005, 2011, and 2016 surveys, respectively. An enumeration area is a primary sampling unit from which households and study participants were recruited. Then, the data collection period from study participants was conducted from April to August 2005, December 2010 to June 2011, and January to June 2016, respectively, for the 2005, 2011, and 2016 reports. The data collection period refers to the time when study participants were recruited. The EDHS data is particularly suitable for health equity analysis due to it consists of the socioeconomic measurement index and other relevant variables [16]. In Ethiopia, household survey data collection was began in 2000 with the first EDHS program, which collected data on knowledge about AIDS. However, it did not include a similar knowledge measurement indicator as the subsequent surveys conducted in 2005, 2011, and 2016. Consequently, we excluded the EDHS 2000 data from the trend analysis.

### Outcome and independent variables

Comprehensive knowledge about HIV/AIDS was the outcome variable. A series of questions were used to generate the level of knowledge among adults who had ever heard of HIV/AIDS. These questions included knowing about the two most common methods to prevent HIV/AIDS infection ('consistent condom use' and 'having one uninfected sexual partner'), providing the correct answer to the question 'Can a healthy-looking person have HIV/AIDS?' and rejecting the two misconceptions about HIV/AIDS ('a person can get HIV from a mosquito bite' and 'a person can get HIV by sharing a meal with people living with HIV'). Each question has three alternatives to be answered, including: yes (coded as 1), no (coded as 0), and do not know (also coded as 0). Those who answered yes to each question were considered to have comprehensive knowledge about HIV/AIDS [16, 35, 36].

Socioeconomic and demographic variables were independent variables in the study. Socioeconomic status or living standards can be assessed using direct approaches (income, expenditure, and consumption) and proxy measures (e.g., an asset index) [37]. EDHS has been using a proxy measure (the wealth index). Variables used to estimate the wealth index were household ownership of consumer goods, household characteristics, drinking water source, and toilet facilities. The wealth index score can be calculated using principal component analysis [38], and EDHS staffs calculated it. The continuous scale of the wealth index was then categorised into poorest, poor, middle, rich, and richest quantile groups. Other variables were age in years,

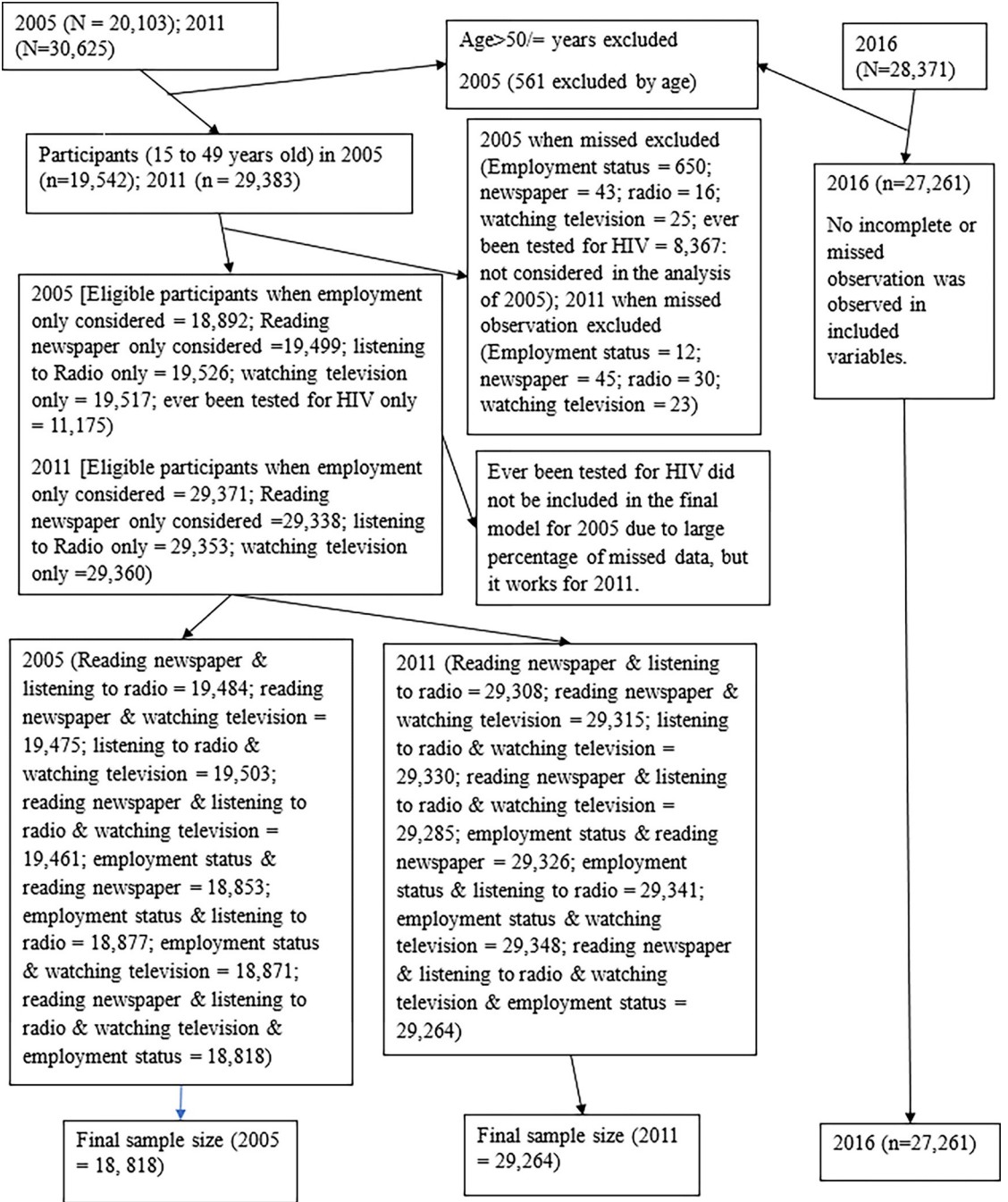

**Fig 1. Displays the flow chart of sample for the analysis.**

gender, place of resident (urban and rural), education status (no education, primary, secondary, and higher education), employment status (employed versus not employed), region (nine geographic regions: Tigray, Amhara, Oromia, Southern Nation Nationalities and People region, Afar, Somali, Benshangul-Gumuz, Harari and Gambella, and two city administrations: Addis Ababa and Dire Dawa), religion (Orthodox, Catholic, Protestant, Muslim, Traditional and others), current marital status (never married/union, married/ living with partner, and

widowed/divorced/no longer living together/separated), media exposure (reading newspaper/magazine, listening to the radio and watching television). Ever been tested for HIV was also included as a health care-related variable [16, 35, 36].

## Data quality assurance

To ensure the quality of the data, EDHS accomplished well-organized fieldwork, involving a supervisor, a field editor, interviewers, biomarker technicians, and a driver. Training and ongoing supervision were provided for the fieldwork team. Additionally, data quality was assured by using standardized and translated tools into several local languages, supervising data collectors using technology monitors, and employing appropriate software for data entry. All procedures contributed to minimizing the risk of bias. The detailed method for each EDHS is available elsewhere [16, 35, 36].

## Statistical analysis

Missing data were managed using the missing completely at random technique [39]. The multistage survey design and sampling weights were considered in the descriptive and analytical results because the EDHS data exhibits a hierarchical nature. The clustering effect (residence and region) was adjusted using 'svy' command in STATA Version 17 (Stata Corp LLC, College Station, TX, the USA 2021), by which all analyses were conducted. All frequency distribution and advanced analysis results were weighted estimates. A percentage was calculated for each measurement indicator of outcome and exploratory variables. Findings were presented in tables and figures. A concentration curve (CC), Erreygers' concentration index (ECI), and decomposition of the ECI were performed to see socioeconomic inequality and contributors. A CC displays the share of comprehensive knowledge about HIV/AIDS accounted for by the cumulative proportions of individuals in the population ranked from poorest to richest. The CC below the equality line denotes health services concentrated on the richest group, and the reverse is true. The "glcurve" statistical command was employed to generate CC [40]. Subsequently, we analysed Erreygers' concentration index (ECI) to see the degree of socioeconomic inequality. ECI is a common statistical method to assess health care inequality when dependent variable has binary outcomes [41]. The ECI is understood that it is twice the area between CC and the line of equality. The value of ECI is between −1 and 1; the exact value of −1 and 1 denotes absolute inequality, and the zero value represents equitable service distribution. A negative value indicates absolute inequality when a disproportionate concentration of comprehensive knowledge about HIV/AIDS among the poor, and a positive value denotes the reverse. It is estimated from the covariance between comprehensive knowledge about HIV/AIDS, and the fractional rank of the study participant by wealth index. 'Conindex' command was used to estimate ECI [42]. The decomposition of the ECI identified contributors to socioeconomic inequality in a comprehensive knowledge about HIV/AIDS [43] using the generalised linear model (GLM) with a binomial distribution and a logit link function. GLM is an approach commonly used to decompose health variables with binary outcomes, serving as a nonlinear regression model [44]. Other studies have also utilized data sets from demographic health surveys [20, 45].

## Ethical considerations

Ethical approval was obtained from Demographic Health Survey (https://dhsprogram.com/) (S1 File). The University of Queensland Institutional Ethical Review Board also exempted the ethical issue of this research (approval project number: 2022/HE001760). Other ethical issues, like consent and confidentiality, such as informed consent from parents for data obtained

from children, were managed by the EDHS data collection team during the data collection period. The EDHS data does not allow for the potential identification of any household or individual in the data file. In EDHS surveys, strict rules were imposed at various steps during survey implementation to prevent the direct or indirect disclosure individual respondents' identities, their households, or clusters. Cluster identification codes are scrambled to prevent potential disclosure. All information identifying a cluster, household, or individual was omitted from the data file. The sampling variables in the data file were sufficient to describe the data structure and allow users to uniquely identify the clusters and households for data analysis. Therefore, the authors did not have access to individual identifiers' information.

## Results

### Characteristics, HIV/AIDS awareness and comprehensive knowledge about HIV/AIDS

Originally, adolescents and adults aged from 15 to 49 years old were 19541 in 2005, 29383 in 2011, and 27261 in 2016. However, after managing incomplete data in some variables, the final sample size became 18,818 for 2005, 29,264 for 2011 and no incomplete response in 2016 (no change in sample size). A weighted percentage of 91.7% of adults had ever heard of HIV/AIDS in 2005 which increased to 97.5% in 2011 and 95.0% in 2016. Proportion of adults aged 15 to 49 years having comprehensive knowledge about HIV/AIDS was 19.8% (95% CI: 18.6%-21.0%) in 2005, 25.9% (95% CI: 24.5%-27.4%) in 2011 and 27.9% (95% CI: 26.5%-29.3%) in 2016 (Table 1).

The frequency distribution of study participants corresponding to the percentage of comprehensive knowledge about HIV/AIDS is presented in Table 2. For example, in 2005, men participants accounted for 28.9%, while in 2011, they accounted for 43,7%, and in 2016, they accounted for 42,5%. The majority of participants in 2005 (82.7%), 2011 (76.7%), and 2016 (78.8%) were rural residents. In terms of employment, approximately 50.6% in 2005, 74.0% in 2011, and 68% in 2016 were employed. Additionally, 5.8% in 2005, 39.8% in 2011, and 44.8% in 2016 were responded that they had ever been tested for HIV (Table 2).

### Wealth-related inequality in comprehensive knowledge about HIV/AIDS

**Concentration curve.** The concentration curve shows higher comprehensive knowledge about HIV/AIDS are concentrated among the richest in 2005, 2011, and 2016 (Fig 2).

**Table 1. HIV/AIDS awareness, knowledge-specific questions, proportion of adults aged 15 to 49 years comprehensive knowledge about HIV/AIDS from 2005 to 2016; percentage distribution is weighted.**

| HIV/AIDS awareness and comprehensive knowledge about HIV/AIDS | 2005 (n = 18,818) | 2011 (n = 29,264) | 2016 (n = 27,261) |
|---|---|---|---|
| Ever heard of AIDS (Yes) | 92.0% | 97.5% | 95.0% |
| Knowledge-specific questions (For those who answered yes for ever heard of AIDS) | n = 17,075 | n = 28,478 | n = 25,542 |
| 1.Reduce risk of getting HIV by always using condoms during sex (Yes) | 51.4% | 68.9% | 69.4% |
| 2. Reduce risk of getting HIV by having only one sex partner with no other partner (Yes) | 73.3% | 70.5% | 77.9% |
| 3. Can a healthy-looking person have HIV (Yes) | 61.2% | 71.7% | 70.6% |
| 4. Can get HIV from mosquito bites (No) | 54.4% | 58.4% | 72.1% |
| 5. Can get HIV by sharing food with person who has AIDS (No) | 74.6% | 81.5% | 87.0% |
| Comprehensive knowledge about HIV/AIDS (answer yes for 1, 2, 3, and no for 4 and 5) | 19.8% (95% CI: 18.6% to 21.0%) | 25.9% (95% CI: 24.5%–27.4%) | 27.9% (95% CI: 26.5 to 29.3%) |

**Table 2. Study participants' characteristics, comprehensive knowledge about HIV/AIDS between sociodemographic groups among adults aged 15 to 49 years using the 2005, 2011 and 2016 Ethiopian demographic.**

| Variables | 2005 (n = 18,818) | | 2011 (n = 29,264) | | 2016 (n = 27,261) | |
|---|---|---|---|---|---|---|
| | Study participants (%) | Comprehensive knowledge (%) | Study participants (%) | Comprehensive knowledge (%) | Study participants (%) | Comprehensive knowledge (%) |
| Overall | | 19.8 | | 25.9 | | 27.9 |
| **Age** | | | | | | |
| 15–19 | 4,378 (23.26) | 24.3 | 6,992 (23.89) | 28.7 | 5,947 (21.81) | 29.9 |
| 20–24 | 3,491 (18.55) | 24.1 | 5,239 (17.90) | 31.3 | 4,640 (17.02) | 31.3 |
| 25–29 | 3,125 (16.60) | 17.0 | 5,436 (18.58) | 27.7 | 4,929 (18.08) | 28.2 |
| 30–34 | 2,494 (13.26) | 15.3 | 3,529 (12.06) | 23.5 | 3,976 (14.58) | 26.5 |
| 35–39 | 2,168 (11.52) | 16.7 | 3,544 (12.11) | 20.9 | 3,314 (12.16) | 25.9 |
| 40–44 | 1,629 (8.66) | 18.1 | 2,373 (8.11) | 20.8 | 2,493 (9.15) | 23.2 |
| 45–49 | 1,533 (8.15) | 15.5 | 2,150 (7.35) | 17.6 | 1,962 (7.20) | 24.8 |
| **Sex** | | | | | | |
| Men | 5,431 (28.86) | 30.0 | 12,796 (43.72) | 34.0 | 11,594 (42.53) | 38.3 |
| Women | 13,387 (71.14) | 15.8 | 16,468 (56.28) | 19.7 | 15,667 (57.47) | 20.2 |
| **Residence** | | | | | | |
| Urban | 3,256 (17.30) | 46.0 | 6,818 (23.30) | 41.6 | 5,774 (21.18) | 43.08 |
| Rural | 15,562 (82.70) | 14.3 | 22.446 (76.70) | 21.2 | 21,487 (78.82) | 23.79 |
| **Region** | | | | | | |
| Tigray | 1,191 (6.33) | 19.1 | 1,874 (6.40) | 29.5 | 1,836 (6.73) | 32.0 |
| Afar | 196 (1.04) | 15.1 | 246 (0.84) | 14.2 | 210 (0.77) | 20.1 |
| Amhara | 4,684 (24.89) | 22.5 | 7,888 (26.95) | 22.8 | 6,622 (24.29) | 31.7 |
| Oromia | 6,797 (36.12) | 17.3 | 10,920 (37.32) | 24.9 | 10,099 (37.05) | 25.1 |
| Somali | 584 (3.10) | 5.1 | 572 (1.96) | 7.1 | 759 (2.79) | 6.9 |
| Benshangul-Gumuz | 166 (0.88) | 17.2 | 312 (1.07) | 26.2 | 278 (1.02) | 21.2 |
| SNNPR | 3,984 (21.17) | 15.5 | 5.534 (18.91) | 29.9 | 5,653 (20.74) | 25.2 |
| Gambela | 59 (0.32) | 12.7 | 128 (0.44) | 26.6 | 78 (0.29) | 31.2 |
| Harari | 52 (0.28) | 35.2 | 89 (0.30) | 25.8 | 67 (0.25) | 26.4 |
| Addis Ababa | 1,009 (5.36) | 30.9 | 1,579 (5.40) | 34 | 1,502 (5.51) | 31.3 |
| Dire Dawa | 96 (0.51) | 50.9 | 122 (0.42) | 38.6 | 157 (0.57) | 46.9 |
| **Education status** | | | | | | |
| No education | 10,979 (58.34) | 9.3 | 12,154 (41.53) | 11.3 | 10,690 (39.21) | 15.2 |
| Primary | 5,065 (26.92) | 23.9 | 13,047 (44.59) | 30.7 | 11,087 (40.67) | 29.8 |
| Secondary education | 2,448 (13.01) | 53.3 | 2,399 (8.20) | 50.4 | 3,599 (13.20) | 43.7 |
| Higher education | 326 (1.73) | 62.5 | 1,664 (5.68) | 59.7 | 1,885 (6.92) | 58.1 |
| **Marital status** | | | | | | |
| Never married | 5,716 (30.38) | 29.3 | 10,033 (34.28) | 34.7 | 8,909 (32.68) | 35.6 |
| Married/living with partner | 11,495 (61.09) | 15.3 | 17,110 (58.47) | 21.8 | 16,648 (61.07) | 24.1 |
| Widowed/divorced/no longer living together/separated | 1,607 (8.54) | 17.7 | 2121 (7.25) | 18.1 | 1,704 (6.25) | 24.1 |
| **Religion** | | | | | | |
| Orthodox | 9,297 (49.40) | 25.1 | 13,949 (47.66) | 28.7 | 11,934 (43.78) | 33.2 |
| Catholic | 215 (1.14) | 18.9 | 298 (1.02) | 29.4 | 197 (0.72) | 21.3 |
| Protestant | 3,504 (18.62) | 17.1 | 6,088 (20.80) | 28.5 | 6,228 (22.85) | 25.8 |
| Muslim | 5,346 (28.41) | 13.1 | 8,361 (28.57) | 19.7 | 8,534 (31.30) | 22.7 |
| Traditional and non-specified | 456 (2.42) | 12.0 | 568 (1.94) | 22.0 | 368 (1.35) | 15.2 |
| **Employment status** | | | | | | |
| Not employed | 9,299 (49.41) | 16.8 | 7,619 (26.03) | 20.1 | 8,737 (32.05) | 19.0 |

*(Continued)*

**Table 2.** (Continued)

| Variables | 2005 (n = 18,818) | | 2011 (n = 29,264) | | 2016 (n = 27,261) | |
|---|---|---|---|---|---|---|
| | Study participants (%) | Comprehensive knowledge (%) | Study participants (%) | Comprehensive knowledge (%) | Study participants (%) | Comprehensive knowledge (%) |
| Employed | 9,519 (50.59) | 22.8 | 21,645 (73.97) | 28.0 | 18,524 (67.95) | 32.1 |
| **Wealth Index** | | | | | | |
| Poorest | 3,239 (17.21) | 9.5 | 5,106 (17.45) | 14.2 | 4,468 (16.39) | 17.7 |
| Poorer | 3,548 (18.85) | 12.5 | 5,393 (18.43) | 17.5 | 4,9223 (18.06) | 22.1 |
| Middle | 3,566 (18.95) | 13.6 | 5,470 (18.69) | 21.7 | 5,219 (19.15) | 23.6 |
| Richer | 3,637 (19.33) | 16.1 | 5,877 (20.08) | 27.1 | 5,560 (20.40) | 26.8 |
| Richest | 4,828 (25.65) | 39.4 | 7,418 (25.35) | 42.4 | 7,091 (26.01) | 42.3 |
| **Reading newspaper** | | | | | | |
| No | 14,977 (79.59) | 13.3 | 21,224 (72.52) | 18.7 | 21,909 (80.37) | 23.7 |
| Yes | 3,841 (20.41) | 45.3 | 8,040 (27.48) | 45.1 | 5,352 (19.63) | 45.1 |
| **Listening to radio** | | | | | | |
| No | 9,621 (51.13) | 10.5 | 10,506 (35.90) | 15.1 | 16,065 (58.93) | 22.1 |
| Yes | 9,197 (48.87) | 29.5 | 18,758 (64.10) | 32.0 | 11,196 (41.07) | 36.1 |
| **Watching television** | | | | | | |
| No | 14,458 (76.83) | 13.3 | 14,227 (48.62) | 16.4 | 17,262 (63.32) | 19.7 |
| Yes | 4,360 (23.17) | 41.5 | 15,037 (51.38) | 35.0 | 9,999 (36.68) | 42.0 |
| **Sex of household head** | | | | | | |
| Male | 15,261 (81.10) | 18.7 | 23,595 (80.63) | 25.6 | 22,009 (80.74) | 27.2 |
| Female | 3,557 (18.90) | 24.5 | 5,669 (19.37) | 27.4 | 5,252 (19.26) | 19.3 |
| **Ever been tested for HIV** (n = 10,858 for 2005) | | | | | | |
| No | 10,226 (94.18) | 21.6 | 17,614 (60.19) | 19.1 | 15,058 (55.24) | 22.1 |
| Yes | 632 (5.82) | 48.0 | 11,650 (39.81) | 36.3 | 12,202 (44.76) | 35.0 |

**Concentration index.** The magnitude of ECI was positive in each year (p<0.001), with values of 0.251, 0.239 and 0.201 in 2005, 2011 and 2016, respectively. These findings indicate that individuals with higher socioeconomic status possess higher comprehensive knowledge about HIV/AIDS than people with low socioeconomic status. This inequality is declined over time with 0.005% per year (r = -0.94; slope = -0.005) (Fig 3).

## Decomposition of concentration index

Decomposition analysis provided the marginal effect, elasticity, ECI and contribution of covariates to the socioeconomic inequality in comprehensive knowledge about HIV/AIDS from 2005 to 2016.

The probability of men having better comprehensive knowledge about HIV/AIDS than women increased from 2005 to 2016. Regarding geographical region, adults living in Somali region (12.3 percentage point in 2005, 11.4 percentage points in 2011 and 23.4 percentage points in 2016) were lower than in Addis Ababa. Adults in the Amhara region were more likely to have comprehensive knowledge in 2005 (5.5 percentage points higher), in 2011 (7.8 percentage points higher) and in 2016 (3.8 percentage points higher) than in Addis Ababa. Education influences comprehensive knowledge about HIV/AIDS in adults in 2005, 2011, and 2016. People attending primary (9.9 percentage points), secondary (15.8 percentage points), and higher (21.9 percentage points) education were more likely to have comprehensive knowledge about HIV/AIDS than non-educated people in 2016. The richest people were more likely to have comprehensive knowledge about HIV/AIDS by 4.9 percentage points in 2005, 8.2 percentage

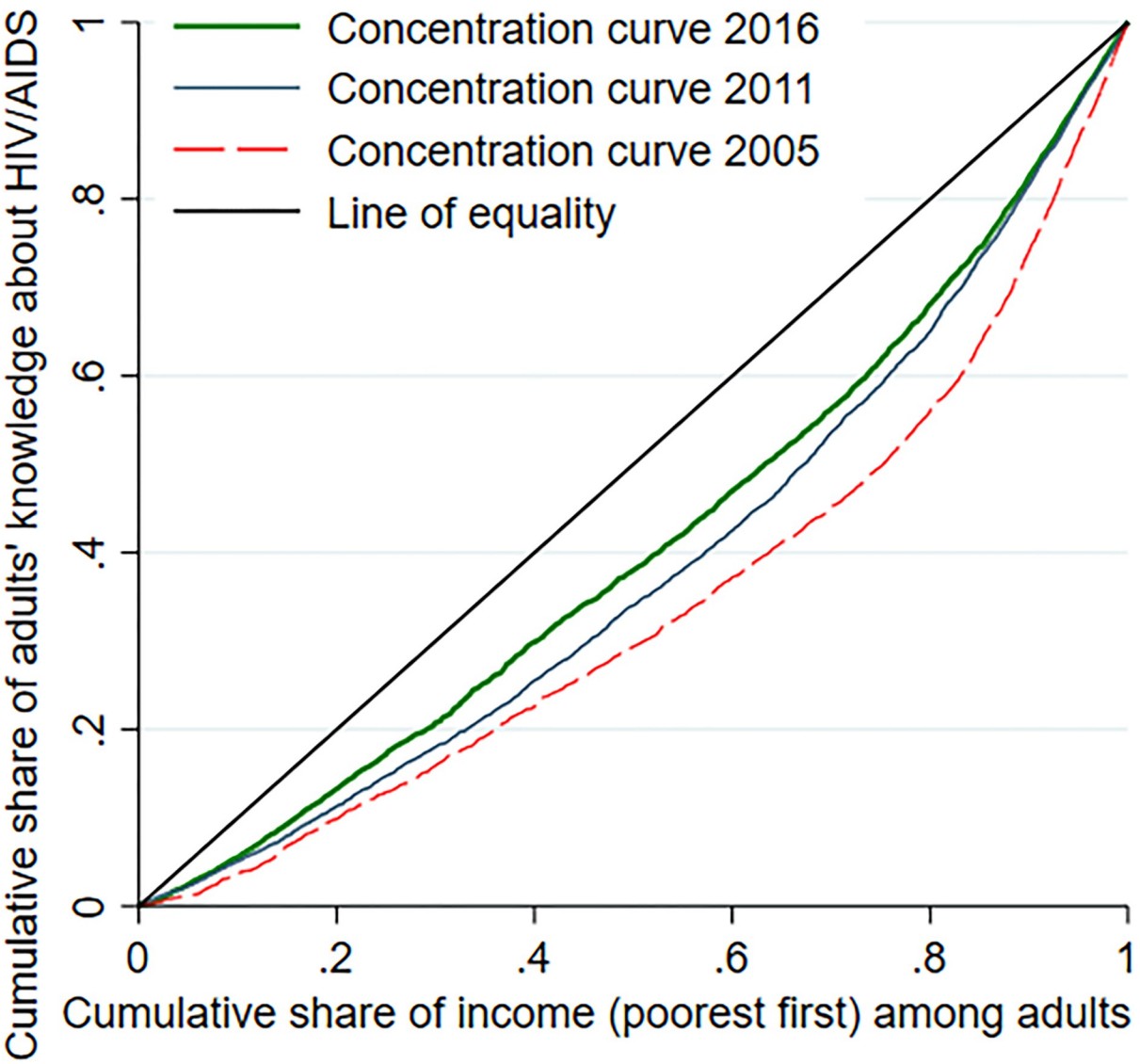

**Fig 2. The CC plots the cumulative percentage of the comprehensive knowledge about HIV/AIDS on the y-axis against the cumulative percentage of the population ranked by living standards (wealth index), beginning with the poorest and ending with the richest on the x-axis.**

points in 2011 and 5.5 percentage points in 2016. Mass-media exposure, specifically reading newspapers and watching television, and having ever been tested for HIV, consistently demonstrate a strong positive effect on comprehensive knowledge about HIV/AIDS. On the other hand, Marital status (never married), religion, region (Amhara and Somali), employment status, and sex of household were the emerged determinants, while the disparities between age, residence, region (Tigray, Oromia, SNNPR), marital status (widowed/separated), religion (Muslim), income (middle, richer), and listening to the radio could not be persisted determinants (Table 3).

The elasticity value for each variable shows a percentage change that a study participant can experience if a participant is part of a reference group. Determinants with a positive elasticity were positively associated with the socioeconomic inequalities in comprehensive knowledge about HIV/AIDS. For instance, the value of elasticity for primary education was 0.159,

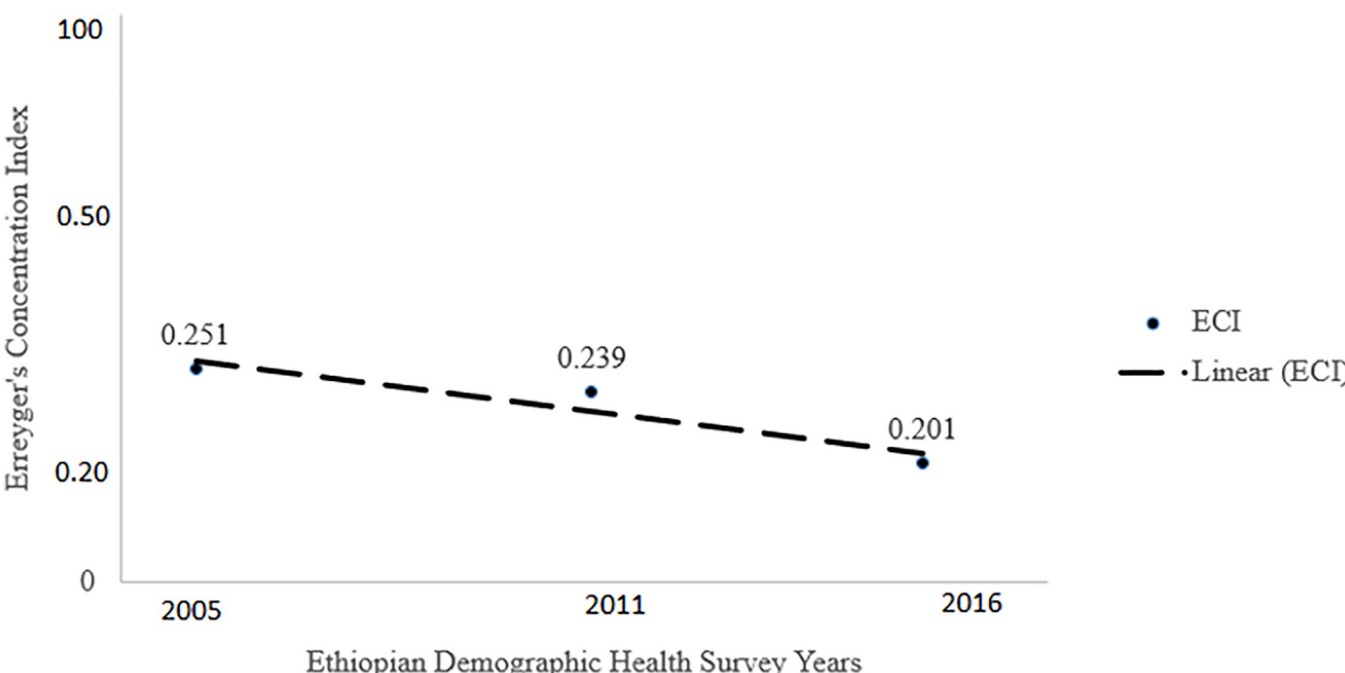

**Fig 3. Trend of socioeconomic inequality in comprehensive knowledge about HIV/AIDS among adults aged 15 to 49 years between 2005 and 2016 in Ethiopia.**

indicating that a change in education status from non-educated to primary education status would result in a 15.9% increment in pro-rich socioeconomic inequality of comprehensive knowledge about HIV. Determinants with a negative elasticity value represents a shift from the reference group to the counterpart, resulting in a reduced pro-rich socioeconomic inequality in comprehensive knowledge about HIV/AIDS.

The ECI of comprehensive knowledge about HIV/AIDS for each covariate indicates the distribution of each specific variable across socioeconomic status. Variables with positive ECI contributed comprehensive knowledge about HIV/AIDS inequality in favouring the rich while those had negative ECI contributed inequality in favouring the poor in comprehensive knowledge about HIV/AIDS. For example, secondary education (0.214), higher education (0.197), never married (0.180), ever been tested for HIV (0.339), employed (0.056), reading newspapers (0.286), watching television (0.507), women household head (0.085) had positive ECI while man (-0.003) and primary education (-0.005) had negative ECI among adults. Major contributors to the socioeconomic inequality in comprehensive knowledge about HIV/AIDS were watching television, household wealth rank, education status, listening to the radio, residence, reading newspaper and geographic region. The percentage contribution of watching television increased from 4.3% in 2005 to 24.2% among both sexes in 2016. Wealth quantile contribution increased from 14.6% in 2005 to 21.38% in 2016. Education status contribution decreased from 16.2% to 14.3%. The percentage contribution of listening to the radio decreased from 16.9% in 2005 to -2.4% in 2016. The percentage contribution of residence decreased from 7.8% in 2005 to -0.5% in 2016 (Table 4 and Fig 4).

## Discussion

Significant socioeconomic inequality was observed in comprehensive knowledge about HIV/AIDS, which gradually declined between 2005 and 2016. Household wealth rank, education

**Table 3. The marginal effects (β) with 95% CI for determinants of comprehensive knowledge about HIV/AIDS among adults aged15 to 49 years from 2005 to 2016 in Ethiopia.**

| Variables | 2005 β (95% CI) | 2011 β (95% CI) | 2016 β (95% CI) |
|---|---|---|---|
| **Age (reference: 15–19 years)** | | | |
| 20–24 | 0. 007 (-0.008, 0.023) | 0.005 (-0.010, 0.020) | -0.001 (-0.018, 0.016) |
| 25–29 | -0.022 (-0.042, -0.003)* | 0.020 (0.003, 0.038)* | -0.013 (-0.033, 0.006) |
| 30–34 | -0.035 (-0.057, -0.013)** | -0.004 (-0.025, 0.016) | 0.008 (-0.013, 0.030) |
| 35–39 | -0.010 (-0.033, 0.013) | -0.023 (-0.044, -0.002)* | 0.017 (-0.006, 0.039) |
| 40–44 | 0.011 (-0.014, 0.036)* | -0.013 (-0.036, 0.011) | -0.023 (-0.048, 0.002) |
| 45–49 | 0.005 (-0.021, 0.031) | -0.023 (-0.049, 0.002) | 0.006 (-0.021, 0.032) |
| **Sex (reference: women)** | | | |
| Man | 0.074 (0.061, 0.086)*** | 0.079 (0.068, 0.089)*** | 0.117 (0.106, 0.129)*** |
| **Residence (Reference: Rural)** | | | |
| Urban | 0.052 (0.033, 0.071)*** | 0.026 (0.007, 0.044)** | -0.002 (-0.022, 0.018) |
| **Region (Reference: Addis Ababa)** | | | |
| Tigray | -0.037 (-0.066, -0.009)* | 0.082 (0.056, 0.108)*** | 0.015 (-0.013, 0.043) |
| Afar | -0.005 (-0.065, 0.054) | -0.007 (-0.074, 0.059) | -0.049 (-0.116, 0.019) |
| Amhara | 0.055 (0.031, 0.078)*** | 0.078 (0.057, 0.100)*** | 0.038 (0.014, 0.061)** |
| Oromia | -0.031 (-0.053, -0.010)** | 0.071 (0.050, 0.092)*** | -0.007 (-0.031, 0.016) |
| Somali | -0.123 (-0.174, -0.072)*** | -0.114 (-0.172, -0.057)*** | -0.234 (-0.290,-0.178)*** |
| Benishangul-Gumuz | 0.009 (-0.050, 0.068) | 0.110 (0.061, 0.159)*** | -0.059 (-0.117, -0.002)* |
| SNNPR | -0.024 (-0.049, 0.001) | 0.116 (0.093, 0.139)*** | 0.005 (-0.021, 0.032) |
| Gambela | -0.081 (-0.186, 0.024) | 0.026 (-0.046, 0.100) | -0.026 (-0.118, 0.066) |
| Harari | -0.025 (-0.108, 0.058) | -0.031 (-0.116, 0.054) | -0.078 (-0.181, 0.024) |
| Dire Dawa | -0.043 (-0.107, 0.022) | 0.049 (-0.020, 0.118) | -0.048 (-0.114, 0.018) |
| **Education (Reference: no education)** | | | |
| Primary | 0.098 (0.083, 0.112)*** | 0.116 (0.103, 0.128)*** | 0.099 (0.085, 0.113)*** |
| Secondary | 0.187 (0.166, 0.203)*** | 0.180 (0.160, 0.199)*** | 0.158 (0.140, 0.177)*** |
| Higher | 0.186 (0.152, 0.221)*** | 0.212 (0.189, 0.235)*** | 0.219 (0.196, 0.242)*** |
| **Marital status (Reference: Married/living with partner)** | | | |
| Never married | 0.008 (-0.024, 0.008) | 0.010 (-0.005, 0.024) | 0.023 (0.007, 0.039)* |
| Widowed/divorced/no longer living together/separated | -0.004 (-0.026, 0.017) | -0.024 (-0.046, -0.003)* | -0.001 (-0.023, 0.022) |
| **Religion (Reference: Orthodox)** | | | |
| Catholic | 0.001 (-0.048, 0.050) | -0.006 (-0.051, 0.040) | -0.071 (-0.134, -0.007)* |
| Protestant | -0.012 (-0.029, 0.005) | -0.008 (-0.022, 0.007) | -0.028 (-0.045, -0.012)*** |
| Muslim | -0.025 (-0.039, -0.010)** | -0.041 (-0.053, -0.028)*** | -0.007 (-0.021, 0.007) |
| Traditional and non-specific | -0.021 (-0.062, 0.019) | -0.020 (-0.057, 0.017) | -0.06 (-0.116, -0.008)* |
| **Employment status (Reference: not employed)** | | | |
| Employed | 0.010 (-0.001, 0.022) | 0.020 (0.007, 0.032)** | 0.034 (0.021, 0.047)*** |
| **Wealth status (Reference: Poorest)** | | | |
| Poorer | 0.008 (-0.012, 0.029) | 0.012 (-0.006, 0.030) | 0.009 (-0.010, 0.028) |
| Middle | 0.001 (-0.020, 0.022) | 0.041 (0.024, 0.059)*** | 0.007 (-0.012, 0.026) |
| Richer | 0.002 (-0.018, 0.023) | 0.052 (0.035, 0.069)*** | 0.012 (-0.006, 0.031) |
| Richest | 0.049 (0.027, 0.072)*** | 0.082 (0.060, 0.104)*** | 0.055 (0.032, 0.077)*** |
| **Reading Newspapers (Reference: No)** | | | |
| Yes | 0.030 (0.016, 0.044)*** | 0.050 (0.038, 0.061)*** | 0.023 (0.010, 0.037)*** |
| **Listening to the radio (Reference: No)** | | | |
| Yes | 0.045 (0.032, 0.057)*** | 0.052 (0.041, 0.064)*** | -0.008 (-0.020, 0.004) |
| **Watching television (Reference: No)** | | | |

(*Continued*)

**Table 3.** (Continued)

| | | | |
|---|---|---|---|
| Yes | 0.025 (0.010, 0.039)*** | 0.029 (0.018, 0.040)*** | 0.065 (0.052, 0.078)*** |
| **Sex of household head** (Reference: Male) | | | |
| Female | 0.009 (-0.005, 0.023) | 0.0002 (-0.012, 0.013) | 0.019 (0.005, 0.032)** |
| **Ever been tested for HIV** (Reference: No) | | | |
| Yes | _ | 0.067 (0.058, 0.077)*** | 0.051 (0.039, 0.062)*** |

* p < 0.05.

** p < 0.01.

*** p < 0.001

status, reading newspapers, watching television, and HIV testing were the largest contributors to socioeconomic inequality in comprehensive knowledge about HIV/AIDS. The urban and rural disparity in wealth-related inequality in comprehensive knowledge about HIV/AIDS declined in 2016. Regarding determinants, sex, regions (Amhara and Somali), education status, richest, mass media (reading newspapers and watching television), and ever being tested for HIV were persistent associated factors.

The current result on the existing socioeconomic disparities is consistent with previous finding in Malawi and Nigeria [20, 46]: people in lower socioeconomic households had lower comprehensive knowledge about HIV/AIDS. This might be due to individuals with lower-income level have limited access to HIV/AIDS prevention information and other services [47]. The lack of information may lead people of low socioeconomic status to make suboptimal decisions regarding health care [48]. Individuals in the poorest household wealth rank may not seek regular health care, resulting in lower levels of knowledge [49]. Other studies among reproductive age women in Ethiopia and sub-Saharan Africa revealed similar findings [17, 28]. In these previous studies, it was found that women living in the poorest household had lower comprehensive knowledge about HIV/AIDS. However, in the current study, over time, despite the significant gap, there has been a decrease in the disparity between the poorest and richest groups decreased as comprehensive knowledge about HIV/AIDS increased among the low-wealth rank group. On the other hand, there has been relatively consistent comprehensive knowledge about HIV/AIDS observed in the highest wealth rank group.

Education status consistently and persistently contributed to socioeconomic inequality in comprehensive knowledge about HIV/AIDS. This result can be explained by individuals in the poorest household wealth rank and those uneducated had lower comprehensive knowledge about HIV/AIDS. A previous study by Chirwa [20] reported similar findings, where individuals with secondary education and above have significantly contributed to socioeconomic inequality compared to their non-educated counterparts. It is understood that higher education status enabled adults' health-seeking behaviors [50]. This leads higher educated individuals to seek connections with health institutions, where they can access health education and counselling services. People who attended health care have higher chance of getting health promotion services. This can be explained by the fact that individuals tested for HIV have shown higher comprehensive knowledge about HIV/AIDS over time, as evidenced by the current study and another comparable study [51].

Mass media exposure was also contributed to the difference in comprehensive knowledge about HIV/AIDS between the richest and poorest economic statuses. Specifically, reading newspaper and watching television resulted in differences in knowledge about HIV/AIDS between poorest and richest economic statuses. Another previous study similarly revealed that reading newspapers significantly contributed to socioeconomic inequality in knowledge about

**Table 4. The elasticity, ECI, and contribution of covariates to the socioeconomic inequality in comprehensive knowledge among adults aged 15 to 49 years in Ethiopia.**

| Variables | 2005 | | | 2011 | | | 2016 | | |
|---|---|---|---|---|---|---|---|---|---|
| | Elasticity | Concentration Index | Percentage contribution Total 65.36% | Elasticity | Concentration Index | Percentage contribution Total 87.82% | Elasticity | Concentration Index | Percentage contribution Total 81.9% |
| **Age (Reference: 15–19 years)** | | | | | | | | | |
| 20–24 | 0.006 | 0.031 | 0.068 | 0.003 | 0.050 | 0.071 | -0.001 | 0.010 | -0.004 |
| 25–29 | -0.015 | -0.0004 | 0.003 | 0.0150 | 0.011 | 0.066 | -0.010 | 0.011 | -0.053 |
| 30–34 | -0.019 | -0.035 | 0.263 | -0.002 | 0.004 | -0.003 | 0.005 | -0.011 | -0.025 |
| 35–39 | -0.005 | -0.027 | 0.051 | -0.011 | -0.018 | 0.080 | 0.008 | -0.015 | -0.062 |
| 40–44 | 0.004 | -0.019 | -0.028 | -0.004 | -0.019 | 0.031 | -0.008 | -0.013 | 0.055 |
| 45–49 | 0.002 | -0.010 | -0.007 | -0.007 | -0.036 | 0.099 | 0.002 | -0.013 | -0.011 |
| Summed | | | 0.35 | | | 0.344 | | | -0.01 |
| **Sex (Reference: Women)** | | | | | | | | | |
| Men | 0.085 | -0.019 | -0.645 | 0.145 | 0.003 | 0.175 | 0.210 | -0.003 | -0.316 |
| **Residence (Reference: Rural)** | | | | | | | | | |
| Urban | 0.036 | 0.547 | 7.82 | 0.024 | 0.643 | 6.35 | -0.002 | 0.601 | -0.510 |
| **Region (Reference: Addis Ababa)** | | | | | | | | | |
| Tigray | -0.009 | -0.041 | 0.154 | 0.021 | -0.010 | -0.091 | 0.004 | -0.022 | -0.044 |
| Afar | -0.0002 | -0.018 | 0.002 | -0.0002 | -0.008 | 0.001 | -0.001 | -0.013 | 0.010 |
| Amhara | 0.054 | -0.045 | -0.975 | 0.085 | -0.114 | -4.08 | 0.038 | -0.015 | -0.269 |
| Oromia | -0.045 | -0.041 | 0.742 | 0.106 | -0.019 | -0.841 | -0.011 | -0.029 | 0.157 |
| Somali | -0.015 | -0.069 | 0.415 | -0.009 | -0.011 | 0.042 | -0.026 | -0.057 | 0.741 |
| Benishangul-Gumuz | 0.0003 | -0.003 | -0.0004 | 0.005 | -0.006 | -0.012 | -0.002 | -0.007 | 0.008 |
| SNNPR | -0.020 | 0.010 | -0.083 | 0.088 | -0.033 | -1.21 | 0.005 | -0.065 | -0.147 |
| Gambela | -0.001 | -0.002 | 0.001 | 0.0004 | 0.003 | 0.0005 | -0.0003 | 0.001 | -0.0002 |
| Harari | -0.0003 | 0.006 | -0.001 | -0.0004 | 0.007 | -0.001 | -0.0008 | 0.004 | -0.002 |
| Dire Dawa | -0.001 | 0.010 | -0.004 | 0.001 | 0.009 | 0.003 | -0.001 | 0.010 | -0.005 |
| Summed | | | 0.251 | | | -6.19 | | | 0.45 |
| **Education (Reference: no education)** | | | | | | | | | |
| Primary | 0.102 | 0.098 | 4.11 | 0.203 | 0.079 | 6.65 | 0.159 | -0.005 | -0.363 |
| Secondary | 0.096 | 0.308 | 11.76 | 0.057 | 0.172 | 4.10 | 0.0818 | 0.214 | 8.69 |
| Higher | 0.013 | 0.060 | 0.306 | 0.048 | 0.166 | 3.29 | 0.061 | 0.197 | 5.99 |
| Summed | | | 16.18 | | | 14.04 | | | 14.32 |
| **Marital status (Reference: Married/living with partner** | | | | | | | | | |
| Never married | -0.010 | 0.195 | -0.742 | 0.013 | 0.173 | 0.937 | 0.029 | 0.180 | 2.60 |
| Widowed/divorced/no longer living together/separated | -0.002 | 0.008 | -0.005 | -0.007 | 0.007 | -0.020 | -0.0002 | 0.007 | -0.001 |
| Summed | | | -0.747 | | | 0.917 | | | 2.60 |
| **Religion (Reference: Orthodox)** | | | | | | | | | |
| Catholic | 0.00003 | 0.003 | 0.00003 | -0.0002 | 0.001 | -0.0001 | -0.002 | 0.002 | -0.002 |
| Protestant | -0.009 | 0.034 | -0.116 | -0.006 | -0.017 | 0.045 | -0.026 | 0.028 | -0.360 |
| Muslim | -0.028 | -0.242 | 2.70 | -0.046 | -0.086 | 1.64 | 0.009 | -0.201 | 0.908 |
| Traditional and non-specified | -0.002 | -0.029 | 0.024 | -0.002 | -0.029 | 0.019 | -0.003 | -0.026 | 0.0426 |
| Summed | | | 2.61 | | | 1.70 | | | 0.59 |

*(Continued)*

**Table 4.** (Continued)

| Variables | 2005 | | | 2011 | | | 2016 | | |
|---|---|---|---|---|---|---|---|---|---|
| | Elasticity | Concentration Index | Percentage contribution Total 65.36% | Elasticity | Concentration Index | Percentage contribution Total 87.82% | Elasticity | Concentration Index | Percentage contribution Total 81.9% |
| **Employment status (Reference: No employed)** | | | | | | | | | |
| Employed | 0.021 | 0.002 | 0.014 | 0.059 | -0.025 | -0.610 | 0.094 | 0.056 | 2.63 |
| **Wealth status** | | | | | | | | | |
| Poorest | Reference | Reference | Reference | Reference | Reference | Reference | Reference | Reference | Reference |
| Poorer | 0.006 | -0.352 | -0.881 | 0.009 | -0.346 | -1.24 | 0.006 | -0.356 | -1.1 |
| Middle | 0.001 | -0.068 | -0.021 | 0.031 | -0.070 | -0.908 | 0.005 | -0.09 | -0.238 |
| Richer | 0.002 | 0.227 | 0.150 | 0.042 | 0.239 | 4.18 | 0.010 | 0.227 | 1.13 |
| Richest | 0.051 | 0.763 | 15.35 | 0.082 | 0.751 | 25.67 | 0.057 | 0.768 | 21.59 |
| Summed | | | 14.60 | | | 27.70 | | | 21.38 |
| **Reading Newspapers (Reference: No)** | | | | | | | | | |
| Yes | 0.025 | 0.365 | 3.56 | 0.054 | 0.362 | 8.22 | 0.018 | 0.286 | 2.61 |
| **Listening to the radio (Reference: No)** | | | | | | | | | |
| Yes | 0.087 | 0.486 | 16.92 | 0.134 | 0.313 | 17.52 | -0.013 | 0.374 | -2.35 |
| **Watching television (Reference: No)** | | | | | | | | | |
| Yes | 0.023 | 0.468 | 4.26 | 0.060 | 0.412 | 10.37 | 0.096 | 0.507 | 24.17 |
| **Sex of household head (Reference: Male)** | | | | | | | | | |
| Female | 0.007 | 0.075 | 0.200 | 0.0001 | 0.086 | 0.004 | 0.014 | 0.085 | 0.582 |
| **Ever been tested for HIV (Reference: No)** | | | | | | | | | |
| Yes | _ | _ | _ | 0.106 | 0.334 | 14.83 | 0.091 | 0.339 | 15.26 |

HIV/AIDS [52]. Mass-media exposure has contributed to socioeconomic inequality in knowledge about HIV/AIDS, and persistently played a role in creating differences between those who have had mass media exposure and those who do not.

Attending HIV/AIDS-related services can improve an individual's comprehensive knowledge about HIV/AIDS. The current study identified that individuals who had ever been tested for HIV had better knowledge. HIV testing provides opportunities for individuals to ask questions, receive detailed information about HIV/AIDS modes of transmission and prevention mechanisms, and gain knowledge measurement indicators [53]. Another study determined that having ever tested for HIV was positively associated with knowledge about HIV/AIDS [54]. It has bidirectional relationship, as knowledgeable individuals are more likely to undergo HIV testing [55]. Therefore, interventions aimed at fostering knowledge about HIV/AIDS may work to facilitate HIV testing provision and increase coverage.

According to the current study, men demonstrated better comprehensive knowledge over time. It is important to note that knowledge about HIV/AIDS can vary based on factors such as education, employment, income, and access to digital technologies [56, 57]. In Ethiopia, men generally have better education and employment opportunity, higher income status, and greater access to the internet compared to women [58–60]. Consequently, men are more likely to access information about HIV/AIDS. Another previous study in Nigeria supported the findings of the current study, indicating that men possessed a higher level of knowledge regarding HIV/AIDS compared to women [46].

An individual and public health approach are required to achieve equality in knowledge about HIV/AIDS. These can be interconnected with a complex system model [61]. Several actors could be engaged in universal HIV/AIDS knowledge creation. For example, school-based HIV/AIDS education, mass-media, and individual or targeted (small group-based)

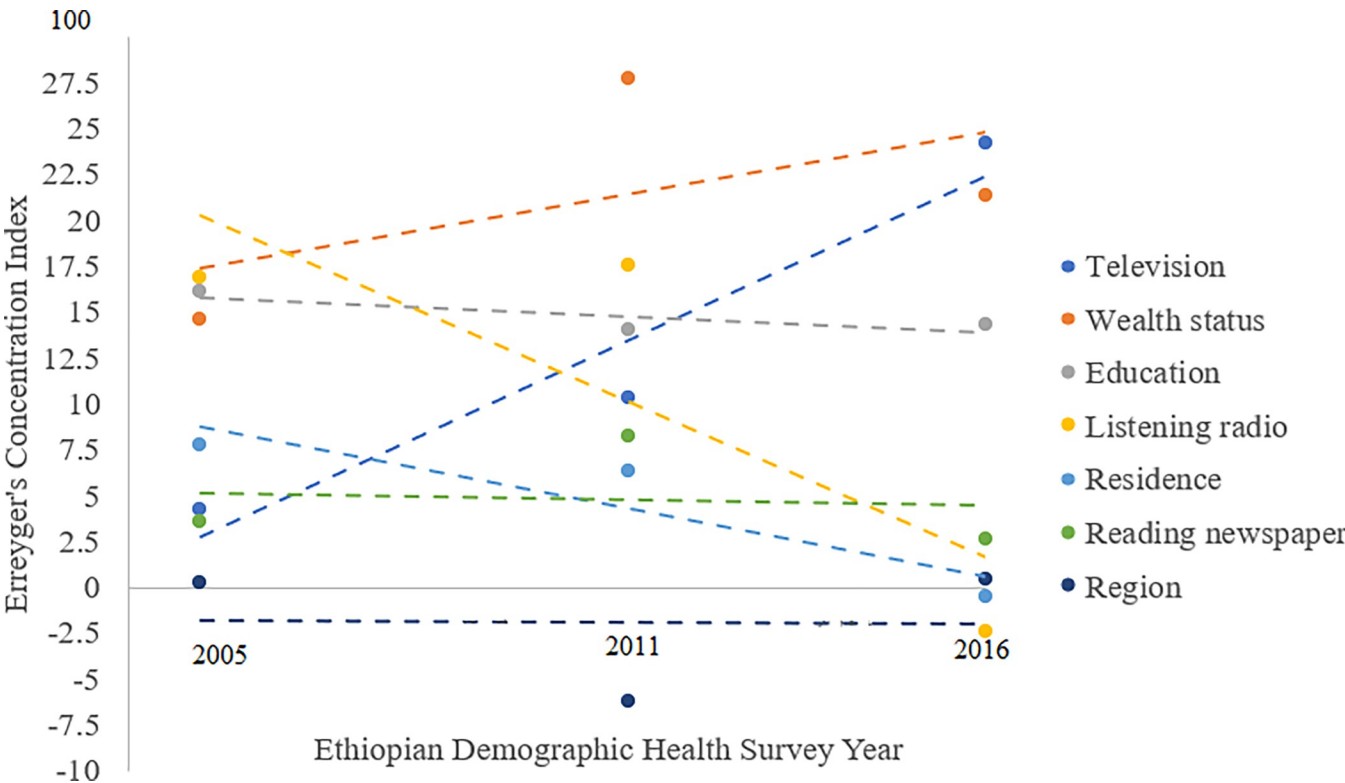

**Fig 4. Percentage contribution of covariates to socioeconomic inequality in knowledge about HIV/AIDS among adults aged 15 to 49 years in Ethiopia over time.**

public health intervention [62, 63]. Recent public health strategy emphasises minimal behavioral change on targeted groups to progressively result in coverage of largest population [64]. The individual-level approach could be highly effective by targeting clients who attend health institutions for different health care services; those who have ever been tested for HIV were more likely to have comprehensive knowledge in this study and other literature [20]. The individual level approach was tested, effectively reducing disparity in other health care services (e.g., smoking cession) [65]. Individual-level interventions, such as cognitive, affective, and behavioral-related have been utilized in the behavioural-related health issues [66]. Some may mention individualised interventions as costly [67]. However, HIV/AIDS interventions could be easily integrated with other house-to-house community-based walk-in services in Ethiopia because many health care services are government funded [68]. Health extension workers have been participating in rural areas at household level, where packages of the rural health extension program include HIV/AIDS prevention and control activities [69]. The World Health Organization also prepared a guide for health policy and system support to engage community health workers to fill the HIV-related knowledge gap in the community [70].

This study has limitations. The parameters of comprehensive knowledge about HIV/AIDS may be prone to recall bias because they were based on respondents' self-report [71]. Disparity changes were based on the oldest and recent EDHS period that does not represent a continuum study favoured from longitudinal data. Longitudinal study is based on longitudinal data, which is collated sequentially from the same respondents repeatedly [72]. Additionally, the distribution of the study participants is a weighted value to each variable in which case some may lose their actual value. Furthermore, the findings from the current study does not exactly show

causation, similar to other priori studies [20]. However, this study's strength of this study lies in its ability to genuinely represent the target population, as the non-response rate was insignificant.

## Conclusions

This study shows comprehensive knowledge about HIV/AIDS was concentrated among those with a higher socioeconomic status. Socioeconomic-related inequality in comprehensive knowledge is woven deeply in Ethiopia, though this disparity has been minimally decreased. A combination of individual and public health approaches entangled in a societal system are crucial remedies for the general population and disadvantaged groups. This requires comprehensive interventions according to the primary health care approach.

## Supporting information

**S1 Checklist. The current study reporting checklists.**
(DOCX)

**S1 File. DHS approval letter: Ethical approval supporting letter.**
(PDF)

## Author Contributions

**Conceptualization:** Aklilu Endalamaw, Yibeltal Assefa.

**Data curation:** Aklilu Endalamaw.

**Formal analysis:** Aklilu Endalamaw.

**Investigation:** Aklilu Endalamaw.

**Methodology:** Aklilu Endalamaw.

**Project administration:** Aklilu Endalamaw.

**Software:** Aklilu Endalamaw.

**Supervision:** Aklilu Endalamaw, Charles F. Gilks, Fentie Ambaw, Yibeltal Assefa.

**Validation:** Aklilu Endalamaw, Charles F. Gilks, Fentie Ambaw, Yibeltal Assefa.

**Visualization:** Aklilu Endalamaw, Resham B. Khatri, Yibeltal Assefa.

**Writing – original draft:** Aklilu Endalamaw.

**Writing – review & editing:** Aklilu Endalamaw, Charles F. Gilks, Fentie Ambaw, Resham B. Khatri, Yibeltal Assefa.

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
