## [Decision Letter · Decision Letter 0]

3 Aug 2023

PGPH-D-23-01054

Socioeconomic inequality in knowledge of HIV/AIDS over time in Ethiopia: a population-based study

Dear Dr. Endalamaw,

Thank you for submitting your manuscript to PLOS Global Public Health. After careful consideration, we feel that it has merit but does not fully meet PLOS Global Public Health’s publication criteria as it currently stands. Therefore, we invite you to submit a revised version of the manuscript that addresses the points raised during the review process.

Editor: Pay particular attention to the concerns and suggestions raised by reviewers #4 and 5. Also complete a thorough grammatical and stylistic revision of the manuscript to ensure clarity and standard English throughout. 

We look forward to receiving your revised manuscript.

Kind regards,

Sarah E. Brewer, PhD

Academic Editor

Journal Requirements:

1. Please provide separate figure files in .tif or .eps format only and remove any figures embedded in your manuscript file. Please also ensure all files are under our size limit of 10MB.

Additional Editor Comments (if provided):

Reviewers' comments:

Reviewer's Responses to Questions

**Comments to the Author**

1. Does this manuscript meet PLOS Global Public Health’s publication criteria? Is the manuscript technically sound, and do the data support the conclusions? The manuscript must describe methodologically and ethically rigorous research with conclusions that are appropriately drawn based on the data presented.

Reviewer #1: Yes

Reviewer #2: Yes

Reviewer #3: Yes

Reviewer #4: Partly

Reviewer #5: Partly

2. Has the statistical analysis been performed appropriately and rigorously?

Reviewer #1: Yes

Reviewer #2: Yes

Reviewer #3: Yes

Reviewer #4: I don't know

Reviewer #5: Yes

3. Have the authors made all data underlying the findings in their manuscript fully available (please refer to the Data Availability Statement at the start of the manuscript PDF file)?

Reviewer #1: Yes

Reviewer #2: Yes

Reviewer #3: Yes

Reviewer #4: No

Reviewer #5: Yes

4. Is the manuscript presented in an intelligible fashion and written in standard English?

Reviewer #1: Yes

Reviewer #2: Yes

Reviewer #3: Yes

Reviewer #4: No

Reviewer #5: Yes

5. Review Comments to the Author

Reviewer #1: • I congratulate the research team for working on HIV knowledge inequality across different variables. As standard concentration index inclusive of corrected indices are directly impacted by sample size and sample weights, I have noted with delight that the researchers have duly ensured that the sample weights were considered in their statistical analytic work.

• Page 4: Methodology—second sentence: revise to read: ‘’we used a cross-sectional design’’.

• Page 18: Under the limitation of the study the last sentence appeared confusing: “Furthermore, the current findings truly represent the target population because response rate was non-significant”. Shouldn’t the statement be referring to ‘’ non-response rate being non-significant rather than response rate’’?

• Align the text in conclusion in the abstract with the conclusion in the main write up on page 18.

Reviewer #2: This manuscript can be published. this is nicely written.

This is a population based survey. It has a great value.

Most of the study does not meet this kind of quality.

The staitsticl part is nicely written based on the objectiveof the study.

Reviewer #3: This is a very good study because if individual are informed about HIV/AIDS and how to prevent or what to do when infected, it is likely they engage in the behavior that will ensure they stay safe. However, knowledge doesn't significantly, alone transform to practice so the author would have tried to check if practices are also difference according to the difference in knowledge. For instance, attitudes towards the use of condoms and early diagnosis rate. Furthermore, it is expected that knowledge and preventive practices will vary across the age group so it is important to try and focus on a particular target group.

Reviewer #4: The topic covered by the article is very salient as it focuses on socioeconomic inequality in comprehensive knowledge on HIV/AIDS over time in Ethiopia. It adds to the body of knowledge on this area and would be a useful resource to health practitioners as they design health programs to improve knowledge on HIV/AIDS.

These are several recommendations to improve the quality of the manuscript.

The article feels rushed and incohesive, with a distinct difference in the writing styles from one section to the next (e.g., use of % then percentage points). The submission would be improved if the authors reviewed grammar, revised and reordered several awkward sentences which hinder clarity, spelt out acronyms when they are first mentioned and reviewed the writing styles in the different sections.

Some points that could be clarified include.

1. The age range of the sample was 15-49 years and is representative of population in that age group. It does not represent the Ethiopians 50 years and older, so not all adults.

2. Was comprehensive knowledge scored 0-5 based on the number of questions answered correctly?

a. How many options were provided for each question?

b. Under results, comprehensive knowledge was given in percentages, what score equates to comprehensive knowledge?

3. What was the Census and Survey Processing System software used? It may not be necessary to include this statement in your manuscript but as it is included, clarification would be useful.

4. Does this study include the limitation that “it does not provide causal evidence about whether the issues outlined are the principal drivers of the observed inequality” (Chirwa et al. 2019). Chirwa GC, Sithole L, Jamu E. Socio-economic Inequality in Comprehensive Knowledge about HIV in Malawi. Malawi Med J. 2019 Jun;31(2):104-111. doi: 10.4314/mmj.v31i2.1. PMID: 31452842; PMCID: PMC6698630.

Recommendations for additions

By including some background on Ethiopia, its population, level of education across its population, development indices and impact of HIV/AIDS in the population, the authors can assist readers to contextualize their research.

Under statistical analysis, referencing other studies which used similar methodologies to analyze demographic health survey days would be useful.

The discussion can be strengthened by elaborating on and comparing the findings to other published studies in Ethiopia and other countries in the region, and expanding on the recent public health strategies being used in Ethiopia.

Reviewer #5: Summary:

The authors have made an excellent effort to find out the demographic and socio-economic factors contributing to the inequality in the knowledge of HIV/AIDS among Ethiopian population. The data for the study was obtained from 2005, 2011, and 2016 population-based health survey. The study concluded that the group with higher incomes had a greater concentration of comprehensive HIV/AIDS knowledge and socioeconomic disparities in comprehensive knowledge of HIV/AIDS have decreased over time.

Comments:

Though the authors have done extensive analysis of the available data, since the authors have done secondary analysis of the already available data source (NHS), I feel it does not fall under the category of original research article. Kindly clarify

Objectives: Needs to be refined further

Study design: Since the data is taken at different time points and compared, can the authors still consider it as a cross-sectional study?

Results: Tables 1, 2 and 3 are too long with too much information in the same table, making it difficult for the readers to comprehend

6. PLOS authors have the option to publish the peer review history of their article (what does this mean?). If published, this will include your full peer review and any attached files.

**Do you want your identity to be public for this peer review?** For information about this choice, including consent withdrawal, please see our Privacy Policy.

Reviewer #1: **Yes: **Dr. Richard AMENYAH

Reviewer #2: **Yes: **Md Abdullah Yusuf

Reviewer #3: **Yes: **Mustapha Adebayo

Reviewer #4: No

Reviewer #5: No

---

## [Decision Letter · Decision Letter 1]

14 Sep 2023

PGPH-D-23-01054R1

Socioeconomic inequality in knowledge about HIV/AIDS over time in Ethiopia: a population-based study

Dear Dr. Endalamaw,

Thank you for submitting your manuscript to PLOS Global Public Health. After careful consideration, we feel that it has merit but does not fully meet PLOS Global Public Health’s publication criteria as it currently stands. Therefore, we invite you to submit a revised version of the manuscript that addresses the points raised during the review process.

EDITOR comments:

Thank you for your thorough revisions of the manuscript.  Reviewer #4 has highlighted a few revisions for consistency (e.g., DHS vs. EDHS) and strength of the discussion that warrant a few more minor revisions.  I hope the authors will choose to make these revisions.  If I am available, I would agree to handle the revised manuscript. 

Please also note that there is disagreement about your data accessibility. Please clearly state how data from this study is accessible or justification for not making data publicly available in your response letter. 

We look forward to receiving your revised manuscript.

Kind regards,

Sarah E. Brewer, PhD

Academic Editor

Journal Requirements:

Additional Editor Comments (if provided):

Reviewers' comments:

Reviewer's Responses to Questions

**Comments to the Author**

1. If the authors have adequately addressed your comments raised in a previous round of review and you feel that this manuscript is now acceptable for publication, you may indicate that here to bypass the “Comments to the Author” section, enter your conflict of interest statement in the “Confidential to Editor” section, and submit your "Accept" recommendation.

Reviewer #1: All comments have been addressed

Reviewer #4: All comments have been addressed

Reviewer #5: All comments have been addressed

2. Does this manuscript meet PLOS Global Public Health’s publication criteria? Is the manuscript technically sound, and do the data support the conclusions? The manuscript must describe methodologically and ethically rigorous research with conclusions that are appropriately drawn based on the data presented.

Reviewer #1: Yes

Reviewer #4: Partly

Reviewer #5: Yes

3. Has the statistical analysis been performed appropriately and rigorously?

Reviewer #1: Yes

Reviewer #4: I don't know

Reviewer #5: Yes

4. Have the authors made all data underlying the findings in their manuscript fully available (please refer to the Data Availability Statement at the start of the manuscript PDF file)?

Reviewer #1: Yes

Reviewer #4: No

Reviewer #5: Yes

5. Is the manuscript presented in an intelligible fashion and written in standard English?

Reviewer #1: Yes

Reviewer #4: Yes

Reviewer #5: Yes

6. Review Comments to the Author

Reviewer #1: Congratulations

Reviewer #4: The manuscript flow has been much improved since your first submission. Congratulations on your hard work to revise the piece and for incorporating the comments of the reviewers.

Minor Revisions Recommended

There are several instances of awkward sentences and incorrect grammar to be fixed.

• For example – “by preventing individuals from engaging in risky sexual behavior”.

Please be consistent with the use of EDHS or DHS.

Cite the source of the information on the EDHS methodology.

Insert a reference for the missing completely at random technique.

Was the method used in the analysis used by other researchers analyzing DHS data? If so, give several examples.

The discussion could be strengthened by elaborating on the findings of some of the studies briefly mentioned.

• Studies revealed similar findings (17, 28).

• Other studies which found that men had better education and employment opportunities, and internet access compared to women (references 57, 58, 59).

• The outcome of the individual level approach (reference 64).

Please clarify the statement in the limitation that this study does not exactly show causation.

Reviewer #5: (No Response)

7. PLOS authors have the option to publish the peer review history of their article (what does this mean?). If published, this will include your full peer review and any attached files.

**Do you want your identity to be public for this peer review?** For information about this choice, including consent withdrawal, please see our Privacy Policy.

Reviewer #1: **Yes: **Richard AMENYAH

Reviewer #4: No

Reviewer #5: No

---

## [Editor Report · Decision Letter 2]

4 Oct 2023

Socioeconomic inequality in knowledge about HIV/AIDS over time in Ethiopia: a population-based study

PGPH-D-23-01054R2

Dear Mr. Endalamaw,

We are pleased to inform you that your manuscript 'Socioeconomic inequality in knowledge about HIV/AIDS over time in Ethiopia: a population-based study' has been provisionally accepted for publication in PLOS Global Public Health.

Best regards,

Sarah E. Brewer, PhD

Academic Editor